# COVID-19 Pandemic-Revealed Consistencies and Inconsistencies in Healthcare: A Medical and Organizational View

**DOI:** 10.3390/healthcare10061018

**Published:** 2022-05-31

**Authors:** Diana Araja, Uldis Berkis, Modra Murovska

**Affiliations:** Institute of Microbiology and Virology, Riga Stradins University, 5 Ratsupites Str., LV-1067 Riga, Latvia; uldis.berkis@rsu.lv (U.B.); modra.murovska@rsu.lv (M.M.)

**Keywords:** healthcare policy, personalized medicine, vaccine-resistance, vaccine-resistant viral variants, myalgic encephalomyelitis/chronic fatigue syndrome (ME/CFS), chronic diseases, people-centered healthcare

## Abstract

The circumstances of the Coronavirus disease caused by the SARS-CoV-2 virus (COVID-19) pandemic have had a significant impact on global and national developments, affecting the existence of society in all its expressions, as well as the lives of people themselves. In the context of the pandemic, increased attention has been focused on acute measures, but the ending of the pandemic is expected as a resolution of the related healthcare problems. However, there are several indicators that the COVID-19 pandemic might induce long-term consequences for individual and public health. Some of the consequences are inferred and predictable, but there are also areas of medicine that have been indirectly affected by the pandemic, and these consequences have not yet been sufficiently explored. This study is focused on drawing attention to some of the COVID-19 pandemic consistencies and the pandemic-revealed inconsistencies in healthcare. Content analysis and statistical analysis were applied to achieve the aim of the study. The main findings of the study address chronic disease burden (particularly, myalgic encephalomyelitis/chronic fatigue syndrome (ME/CFS)), healthcare governance and organizational issues, and the synergy between health policy perspectives and innovative solutions in practice. The study provides insight into the particular healthcare issues affected by the COVID-19 pandemic, such as the increase in mortality in some diagnoses besides COVID-19 and the possible emergence of a new type of resistance—vaccine-resistance—contemporaneously supporting the identification of the tendencies and currently unnoticed indirect consistencies and inconsistencies revealed by the pandemic.

## 1. Introduction

The International Health Regulations (2005) is a legally binding international instrument used to help countries work together to save lives and minimize the impact on livelihoods by events that cause the international spread of diseases [1]. Even before the global financial crisis began in 2008, the World Health Organization (WHO) had stated that, in June 2007, when the revised International Health Regulations entered into force, the world would also have the necessary global framework to prevent, detect, assess, and provide a coordinated response to events that may constitute a public health emergency of international concern [2]. On 12 December 2012, the United Nations General Assembly endorsed the resolution on Global Health and Foreign Policy promoting synergy between foreign policy and global health and urging countries to accelerate progress toward Universal Health Coverage (UHC). The concept anticipated that all individuals and communities should receive the health services they need without suffering financial hardship [3]. All the United Nations Member States anticipate achieving UHC by 2030, as part of the Sustainable Development Goals [4].

The necessity of reimagining healthcare governance was also previously emphasized in the context of emergency situations. In the following years, the WHO continued to call for disaster preparedness and the strengthening of healthcare systems. The WHO and its partners established the WHO Thematic Platform for Health Emergency and Disaster Risk Management Research Network (HEALTH EDRM RN) in 2016 to respond to the increasing burden of recent health emergencies and disasters [5].

One of the most recent documents is the WHO position paper ’Building health systems’ resilience for universal health coverage and health security during the COVID-19 pandemic’, that provides a rationale and recommendations for building resilience and seeking integration between promoting UHC and ensuring health security by the following appropriate means [6]:-Recovery and transformation of national health systems through investment in the essential public health functions and the foundations of the health system, with a focus on the primary healthcare and the incorporation of health security.-All-hazards emergency risk management, to ensure and accelerate sustainable implementation of the International Health Regulations (2005).-Whole-of-government approach to ensure community engagement and whole-of-society involvement.

The framework of this research is dedicated to healthcare-related policies and their implementation in practice, placing emphasis on particular tendencies. Simultaneously, the COVID-19 pandemic, as an accelerator of the lengthy processes, revealed some of the policies’ contrarieties that existed before the pandemic but escalated during it and created the context of the research.

In particular, indications of the divergence between policies and their practical implementation had already emerged, which is significantly associated with the implementation of the precision medicine concept. In accordance with US Food and Drug Administration (FDA), ’precision medicine’ is an innovative approach to tailoring disease prevention and treatment that considers differences in people’s genes, environments, and lifestyles. The goal of precision medicine is to target the right treatments to the right patients at the right time [7]. The European Union’s Health Ministers in their Council conclusions on personalized medicine for patients, published in December 2015, defined personalized medicine as a ’medical model using characterization of individuals’ phenotypes and genotypes (e.g., molecular profiling, medical imaging, lifestyle data)’, additionally noting that personalized medicine should be seen as an evolution of medicine, rather than a revolution, and many challenges remain for its successful application across healthcare systems [8].

The application of the principle of personalized medicine to vaccines has been named vaccinogenomics, or vaccinomics, and this term was first time used in 1998 [9], well before the COVID-19 pandemic. By definition, vaccinomics explores the influence of genetic (and non-genetic) factors on the heterogeneity of vaccine-induced immune responses between individuals and populations. Vaccinomics provides the opportunity to examine not only immune response genes likely to be involved in vaccine response, but also the possibility of identifying the influence of new (uncharacterized) genes on vaccine-induced immunity [10]. While vaccinomics has been more focused on vaccine effectiveness, the field of adversomics is concerned about the side effects of vaccines [11], and its topicality remains in the COVID-19 pandemic. Ten years ago, scientists assumed that vaccinomics and the immune response theory will poise to be the major drivers of how vaccines will be developed, administered, and monitored in the 21st century. In addition, the application of these concepts to adversomics will enable the concept of personalized vaccinology and allow administration of vaccines informed by more refined estimates of potential adverse side effects to a given vaccine. It was advised that “the application of these concepts may result in significantly diminishing the anti-vaccine and general fear of vaccines that seem so prevalent today, and that measurably—and adversely—affect population coverage with vaccines” [12]. At present, ten years later, the implementation of this advice is critically considered.

Regarding SARS-CoV-2 vaccines, the studies identified that personalized approaches to SARS-CoV-2 vaccine development will be even more important than for other vaccines, as profound differences in the patient responses to SARS-CoV-2 infection have been reported since the early stages of the COVID-19 pandemic. Several environmental and demographic factors have been attributed to the interindividual differences in COVID-19 severity, but there are some indications that genetic factors have an important role here. Considering SARS-CoV-2 vaccine development, it should be noted that patients who have severe reactions to the infection might be more at risk for adverse reactions to the vaccine. Conversely, individuals with asymptomatic presentation might be potential vaccine non-responders. Thus, identification of the genetic factors that modulate responses to vaccine administration is of the utmost importance for the development of safe and effective vaccines, to increase the public trust in vaccines, and consequently, to increase the background vaccination rates. [13]

Despite the scientific principles of the personalized vaccinology, during the current pandemic, the SARS-CoV-2 vaccines are used in mass off-site vaccination events, mostly with only one diagnostic manipulation—body temperature measurements, thereby neglecting the principles of precision medicine. The question arises whether the current hierarchical healthcare governance and organizational model, by which the decisions on the administration of medicinal products to people are made on the highest governmental level, far from medicinal opinion, is the most appropriate for the management of advanced technologies in stressful circumstances so that the benefits for society and individuals from their use outweigh the risks, without discreditation of new health technologies in the society’s views.

The contemporary meaning of healthcare governance is based on the three traditional levels of macro-, meso-, and micro-level. Macro-level mostly consists of national-level policy-making functions, meso-level consists of institutional-level decision-making functions, while micro-level concerns operational issues at the clinic or physician office level [14]. The distinction of competencies was rather blurred even before, but mostly between meso- and micro-levels, however now the disorganization of levels becomes more considerable.

Regarding the global healthcare policy-planning documents and reflecting on the real situation, the discrepancy in the theoretical directions and practical performance is apparent. The greatest emphasis in the last two years has been on strengthening health systems by attracting more investment and increasing spending budgets. In Latvia, for instance, the temporary budget expansion in response to COVID-19 has been more than EUR 3.5 billion in the last two years, compared to an average annual budget allocation of EUR 1.3 billion for healthcare [15,16]. However, there are signs that an increase in funding, without proper control over its use, is not a panacea for improving the functioning of healthcare systems, as it does not necessarily lead to a proportionate health services supply increase. Within the limits of this study, the efficiency of resource allocation during the pandemic was not evaluated, but the research results might be indirectly related to the efficiency of resource allocation.

The study hypothesizes that not all pandemic consequences can be predicted and supervised, despite the existence of many policy-planning and healthcare governance documents. The aim of this research is to evaluate some pandemic-revealed consistencies and inconsistencies in healthcare, to help in forecasting future scenarios and increase their value under persistent conditions of uncertainty.

## 2. Materials and Methods

The study was designed to investigate COVID-19 pandemic-revealed consistencies and inconsistencies in two aspects:The medical aspect.The healthcare organization aspect.

The ‘medical aspect’ of the research was dedicated to such consequences of COVID-19 as an increase in the prevalence of chronic diseases, particularly ME/CFS, changes in the mortality rates of diseases not directly related to COVID-19, and immunization tendencies. ME/CFS is a poorly understood, serious, complex, multisystem disorder, characterized by symptoms lasting at least six months, with severe incapacitating fatigue not alleviated by rest, and other symptoms—many autonomic or cognitive in nature—including profound fatigue, cognitive dysfunction, sleep disturbances, muscle pain, and post-exertional malaise, which lead to substantial reductions in the functional activity and quality of life. The prevalence of this disease in the population varies from 0.19% to 7.6% [17].

Regarding the defining post-COVID-19 symptoms, based on the relapsing/remitting nature of post-COVID-19 symptoms, the following integrative classification was proposed: potentially infection-related symptoms (up to 4–5 weeks), acute post-COVID-19 symptoms (from week 5 to week 12), long post-COVID-19 symptoms (from week 12 to week 24), and persistent post-COVID-19 symptoms (lasting more than 24 weeks) [18]. Therefore, in the context of the time reference, ME/CFS is most closely associated with persistent post-COVID symptoms. Persistent post-COVID symptoms were considered in light of a new “syndrome”, as the British Medical Association defines a syndrome “as a set of medical signs and symptoms which are correlated with each other and associated with a particular disease” [18]. Accordingly, the term “persistent post-COVID-19 syndrome (PPCS)” was introduced in practice, as a pathologic entity, which involves persistent physical, medical, and cognitive sequelae following COVID-19, including persistent immunosuppression as well as pulmonary, cardiac, and vascular fibrosis [19].

Meanwhile, the ‘healthcare organization aspect’ is focused on the elaboration of the suitability of the dominant healthcare governance and organizational model for the priorities of contemporary medicine’s development.

The statistical data produced by the WHO (Coronavirus (COVID-19) Dashboard) and the national competent authorities of Latvia (Latvian Centre for Disease Prevention and Control (CDPC), Health Statistic Database; Central Statistical Bureau, Official Statistic of Latvia) were used to make preliminary predictions on some of the COVID-19 consequences, described in Section 3.1.2: ’Mortality rates in indirect COVID-19 related diagnoses’ and Section 3.1.3: ’Immunization tendencies’.

## 3. Results

### 3.1. Medical Aspect

#### 3.1.1. Chronic Disease Burden

In previously conducted research, authors performed a literature review dedicated to the reciprocity of COVID-19 and ME/CFS and have indicated eight scientific articles in the period from December 2019 to March 2021 [20]. Later, researchers emphasized that the studies into long-COVID-19 symptomatology suggest many overlaps with clinical presentation of ME/CFS. Advancements and standardization of long-COVID-19 research methodologies would improve the quality of future research and may allow further investigations into the similarities and differences between long-COVID-19 and ME/CFS [21]. Despite important similarities between post-acute COVID-19 symptoms and ME/CFS, there is currently insufficient evidence to establish COVID-19 as an infectious trigger for ME/CFS. Further research is required to determine the natural history of this condition, as well as to define risk factors, prevalence, and possible interventions [22].

Although most patients recover from acute COVID-19, some experience post-acute sequelae of severe acute respiratory syndrome coronavirus 2 infection (PASC). One subgroup of PASC is a syndrome called ’long-COVID-19’, reminiscent of ME/CFS [23]. Another group of researchers proposed different taxonomy, and PASC as post-acute sequelae of SARS-CoV-2 infection, which include injury to the lungs, heart, kidneys, and brain that may produce a variety of symptoms. In their view, PASC also includes a post-COVID-19 syndrome (‘long-COVID’) with features that can follow other acute infectious diseases and ME/CFS. Researchers summarized that the pathogenesis of post-COVID-19 syndrome in some people may be similar to that of ME/CFS. Researchers proposed molecular mechanisms that might explain the fatigue and related symptoms in both illnesses and suggested a research agenda for both ME/CFS and post-COVID-19 syndrome [24].

There is also the view that despite the similarities between the symptoms of long-COVID patients and ME/CFS, further evidence is required to list COVID-19 among the infections associated with ME/CFS. Additional investigations with longer follow-ups, more uniform criteria for ME/CFS diagnosis, including both in- and out-patients with infections of different severity, and a control group of people affected by other infections are required to better characterize risk factors, prevalence, and progression of long-COVID ME/CFS-like features, and to design specific interventions and treatments [25].

In the systematic review and meta-analysis of 81 studies, researchers established that approximately a third of the included individuals experienced persistent fatigue and over a fifth of individuals exhibited cognitive impairment 12 or more weeks following COVID-19 diagnosis. Furthermore, a subset of individuals exhibited markers of systemic inflammation, and ’post-COVID-19 syndrome’ (PCS) was associated with marked levels of functional impairment. Limited evidence suggested a possible association between elevated pro-inflammatory markers and fatigue or cognitive impairment in PCS. Future research should endeavor to identify the underlying mechanisms, develop standardized diagnostic criteria, and establish therapies to prevent and treat fatigue and cognitive impairment [26]. In view of other researchers, the definition of long-COVID-19 and the establishment of its temporality vary, but some researchers consider it possible that this syndrome is actually ME. Similarities are observed when comparing the International Consensus Criteria for the diagnosis of ME with the symptoms described for long-COVID-19. The possibility that COVID-19 can lead to a chronic condition such as ME makes long-term follow-up of patients who have suffered from this infection essential [27].

The reviews’ results highlight problems associated with the ME/CFS, long-COVID-19, and post-COVID syndrome definitions, and classification criteria [23,24,26]. There are various manifestations of the interaction between COVID-19 and ME/CFS, from the similarity of the symptoms [21,25,26], also with reserved view [22], to the ME/CFS as a consequence of PASC [23,24], and the assumption that it is the same disease [27]. At the same time, all publications indicate the need for further research into the similarities and differences between post-COVID-19 and ME/CFS, to determine the natural history of this condition, as well as to define risk factors, prevalence, and possible interventions [21,22,23,24,25,26,27].

The original research articles are dedicated to specific studies performed in different countries and population groups [28,29,30,31,32,33], with different approaches to selection criteria, key agents, nosology, and definitions [34,35,36,37,38], and different methods of diagnostics and treatment, as well as measuring the patients’ reported outcomes [39,40,41,42,43,44,45,46,47]. The results range from an inconclusive interaction between COVID-19 and ME/CFS to assumptions about the analogy of post-COVID-19 syndrome with ME/CFS. In the overview, considerable attention was paid to the complex symptoms that are characteristic of several diseases, and the new challenges of nosology.

Additionally, the viewpoint article [48] notes that autonomic dysfunction after a viral illness, which has been observed in long-COVID-19, is also a feature of similar conditions such as ME/CFS and can cause disabling syndromes, including postural orthostatic tachycardia syndrome (POTS). Long-COVID-19 research may also be applicable to a wider range of chronic illnesses, including ME/CFS, which similarly lack adequate understanding and recognition [48]. Meanwhile, the short communication provides predictions, as the preliminary findings raise concerns regarding a possible future ME/CFS-like pandemic in SARS-CoV-2 survivors [49].

The pandemic not only causes long-term direct consequences, but also induces conditions for changes in mortality rates besides COVID-19 group of diagnoses.

#### 3.1.2. Mortality Rates in Indirect COVID-19-Related Diagnoses

The COVID-19 pandemic is increasing mortality rates worldwide. Globally, in the beginning of May 2022, there have been almost 6.3 million deaths caused by COVID-19 reported to the WHO [50]. Researchers are paying close attention to assessing these direct effects. However, in the scope of this study, the authors analyzed the possible indirect impact of the pandemic on mortality rates in diagnoses besides COVID-19.

Latvian data were used for the analysis, and three diagnosis groups with the highest mortality rates, according to the Latvian Centre for Disease Prevention and Control, were selected for analysis:(1)Diseases of the circulatory system (I00-I99, ICD-10)(2)Neoplasms (C00-D48, ICD-10)(3)External causes of death (V00-Y98, ICD-10)

Data from 2015 to 2021 were analyzed to determine statistical data trends.

The results of the analysis demonstrate that there were no significant trend changes in the diagnosis groups neoplasms (C00-D48) and external causes of death (V00-Y98). In 2021, mortality caused by COVID-19 (U07.1, U07.2), post-COVID-19 condition (U09), and multisystem inflammatory syndrome associated with COVID-19 (U10) ranked third, ahead of external causes of death. Simultaneously, in the group of diseases of the circulatory system (I00-I99), in which a gradual and steady decrease in mortality was observed from 2015 to 2019, the mortality increased by 2.8% in 2020, as compared with 2019. Meanwhile, 2021 demonstrated extremely high growth—by 9% compared to 2020 [43]. Figure 1 demonstrates the changes in mortality rates of the group of diseases of the circulatory system (I00-I99) from 2015 to 2021.

The example of Latvia shows that during the pandemic, mortality increased notably in some diagnoses besides COVID-19, namely in the group of diseases of the circulatory system (I00-I99) [51]. There is a significant volume of published literature on the increase in excess mortality and other conditions that are related to COVID-19 despite not being included in COVID-19-related morbidity and mortality statistics. Cardiovascular complications from COVID-19 were observed as early as the beginning of the pandemic [52,53,54,55,56]. The most common cardiovascular complications include acute myocardial injury, arrhythmias, acute myocarditis, and severe left ventricular dysfunction [57]. On the other hand, patients with cardiovascular comorbidities are more prone to suffer from COVID-19, which in turn can cause deterioration of their cardiovascular disease [53].

Simultaneously, researchers reported that vaccination with mRNA-1273 was associated with a significantly increased risk of myocarditis or myopericarditis in the Danish population, primarily driven by an increased risk among individuals aged 12–39 years, while BNT162b2 vaccination was only associated with a significantly increased risk among women [58]. Based on passive surveillance reporting in the US, the risk of myocarditis after receiving mRNA-based COVID-19 vaccines was increased across multiple age and sex strata and was highest after the second vaccination dose in adolescent males and young men [59]. Similar results have also been reported in recent (2022) original studies [60,61] and systematic literature reviews [62,63], moreover demonstrating that myocarditis and pericarditis after the COVID-19 vaccine have been reported more in young adult males and are most likely to occur after the second dose of mRNA vaccines [63]. The researchers emphasized that the benefits of SARS-CoV-2 vaccination should be considered when interpreting these findings.

However, the authors assume that the mortality rates in cardiovascular diseases could be induced not only by COVID-19 illness or side effects of vaccines but also due to strictly limited access to primary healthcare services and hospital admissions for cardiac care during pandemic-related lockdowns. In accordance with the observational study performed in Germany, during the COVID-19-related lockdown, a significant increase in cardiovascular mortality was observed in central Germany, whereas catheterization activities were reduced [64]. Research conducted in Lithuania suggests that the state-wide lockdown measures could have had unwanted collateral consequences on prehospital stroke care [65]. A cross-sectional study performed in the UK concluded that individuals with a chronic illness were more likely to experience canceled healthcare appointments and greater care needs during the UK national lockdown generated by the COVID-19 pandemic [66]. In the authors view, the impact of pandemic-related lockdowns induced long-term consequences for public health, and further research will reveal to what extent mortality increases are attributable to healthcare services’ unavailability.

#### 3.1.3. Immunization Tendencies

Immunization was already defined as one of the priorities in policy planning, research, and programming before the COVID-19 pandemic. The importance of immunization was emphasized by the United Nations Children’s Fund (UNICEF) and the WHO in the strategic documents and performed activities. The research was conducted in diverse directions, for instance, the European Network of Vaccine Research and Development (TRANSVAC) was a collaborative infrastructure project aimed at enhancing European translational vaccine research and training. The objective of this four-year project (2009–2013), funded under the European Commission’s seventh framework program (FP7), was to support European collaboration in the vaccine field, principally through the provision of transnational access to critical vaccine research and development infrastructures. The project successfully provided all available services to advance 29 projects and accelerated the development of vaccines for diseases such as Lyme disease, malaria, tuberculosis, dengue, influenza, mumps, whooping cough, pneumonia, HIV, and two types of cancer [67]. The next generations of the TRANSVAC project are continuing this work towards a sustainable European vaccine infrastructure.

At the 2012 World Health Assembly, ministers for health endorsed the Global Vaccine Action Plan, to ensure that by 2020 no one would miss out on vital immunization. In 2014, the European Regional Committee of the WHO adopted the European Vaccine Action Plan 2015–2020. A Joint Action on Vaccination, co-funded by the third program for the European Union’s action in the field of health, starting in 2018, focuses on technical requirements regarding electronic immunization information systems, vaccine forecasting, prioritization of vaccine research, and development and research addressing vaccine hesitancy.

The Council of the European Union, in the Recommendation of 7 December 2018 on strengthened co-operation against vaccine-preventable diseases, noted the necessity of strengthening existing partnerships with international actors and initiatives, such as the WHO and its Strategic Advisory Group of Experts on Immunization (SAGE), the European Technical Advisory Group of Experts on Immunization (ETAGE), UNICEF, and financing and research initiatives such as Gavi (the Vaccine Alliance), CEPI (the Coalition for Epidemic Preparedness Innovations), GloPID-R (the Global Research Collaboration for Infectious Disease Preparedness), and other institutions and organizations [68].

A significant number of policies and planning documents have been developed at the international and national levels, followed by new documents such as the WHO Immunization Agenda 2030: A Global Strategy To Leave No One Behind [69], the WHO European Immunization Agenda 2030 [70], the European Union’s Vaccines Strategy [71], the Vaccines National Strategic Plan for the United States: 2021–2025 [72], and the United States (US) Centers’ for Disease Control and Prevention (CDC’s) Strategy to Reinforce Confidence in COVID-19 Vaccines [73].

Accordingly, objectives have been set, a network of institutions has been set up, research, manufacturing, distribution, and treatment facilities have been engaged, a significant private–public partnership model has been put in place, financial resources have been invested, and communication platforms have been maintained. However, the COVID-19 pandemic shows some inconsistencies between theoretical models and practical implementation (Figure 2 demonstrates the example of Latvia, whose trends are also emerging in other countries).

In Figure 2, the curve colored in red characterizes the proportion of people completely vaccinated against COVID-19 (a completely vaccinated person is a person who has received a full course of vaccination at least 14 days after the completion of the vaccination course with any vaccine registered by the European Medicines Agency or an equivalent regulator or recognized by the WHO) in the total population of Latvia, and the curve colored in blue represents the proportion of people completely vaccinated against COVID-19 in the total number of COVID-19-infected persons in Latvia. A period of more than seven months was considered (from week 41 of 2021 until week 17 of 2022). Significantly, the proportion of people completely vaccinated against COVID-19 in the total number of COVID-19 infected persons exceeded the level of 50% in the last week of 2021, although the number of registered cases of the SARS-CoV-2 Omicron variant was only 164 cases. Subsequently, the curve grew more severely, and a specific leap was observed in the first two weeks of 2022, which was probably influenced by the repealing of all restrictions for vaccinated persons at the turn of the year. The initiation of the booster did not change the tendency of the vaccination curve and it continued to move to the curve ’the proportion of people completely vaccinated against COVID-19 in the total population of Latvia’.

The closer these curves (Figure 2) are, the less the preventive effectiveness and transmission-reducing effect of the vaccines are proven (the effect on the severity of the disease is more permanent). In the eleventh week of 2022, the curves crossed, and this tendency is becoming a serious signal that the SARS-CoV-2 vaccine-resistance should be considered, as a threatening consistency of the pandemic, or as an inconsistency of healthcare policies. Vaccine-resistant viral variants are described in the literature, for instance in [76], however, at least in Latvia, the tendency of increasing infection in vaccinated people (Figure 2) was not dependent on the viral variant.

### 3.2. The Healthcare Organization Aspect

The period in which the pandemic occurred also exacerbated issues related to the organization and governance of healthcare. For many years, healthcare reforms have taken place in virtually every country of the world, the results of which are ambiguous. The main directions of the reforms were focused on the development of primary healthcare, the establishment of a people-centered system, and the optimization of infrastructure and resource allocation. The World Health Report 2008 ’Primary Healthcare’ came 30 years after the Alma-Ata Conference of 1978 on primary healthcare, which agreed to tackle the ’politically, socially, and economically unacceptable’ health inequalities in all countries, and declared that ‘The people have the right and duty to participate individually and collectively in the planning and implementation of their healthcare’ [77]. The following waves of reforms were generated in subsequent years. One of the most recent acknowledgments was the Ministerial Statement ‘The Next Generation of Health Reforms’ of the Organization for Economic Co-operation and Development (OECD), adopted by the OECD Health Ministerial Meeting, on 17 January 2017. The Statement declared the following priorities [78]:Promoting high-value health systems for allAdapting health systems to new technologies and innovationReorienting health systems to become more people-centeredEncouraging dialogue and international co-operation

Concerning the reorientation of health systems towards becoming more people-centered, it was recognized that patients’ expectations are rising, and that there is a need for a fundamental change in how people interact with health services and health professionals. However, as the demand to become active participants in the decision-making is becoming the new normal, people are still too often playing an insufficient role in deciding about their treatment [78]. A review of the progress on this work was planned to take place at the next Health Ministerial meeting in five to six years’ time, i.e., in 2022 or 2023 [79].

Additionally, the COVID-19 pandemic has introduced new adjustments to the implementation of the plans, and the experience of the last two years has demonstrated that people are significantly removed from making decisions about their healthcare. Access to primary healthcare was limited, with negative consequences for the care of patients with chronic diseases, and the workload of hospitals has increased unusually over the last year and a half.

To overview the scientific research on healthcare systems’ issues during the COVID-19 pandemic, the scientific literature was analyzed. The research results demonstrate the main findings provided by the scientists, which can lead to a clear view on the consistencies and inconsistencies in healthcare organization and governance, revealed by the pandemic:Traditional organizational cultures that are hierarchical and controlling need to be challenged and reoriented towards collaborative, inclusive, and participative practices of engagement and involvement. Health systems’ processes must move from a top-down approach to integrative policy, planning, and implementation processes, increasingly adopting a people-centered approach [80,81,82].COVID-19 has brought upon us the opportunity to redesign health policy thinking and acting to stimulate new possibilities through critical debate, moving from fragmentation to adaptive self-organization, creating well-integrated, equitable, and prosperous societies resilient to sudden unexpected perturbations of any kind [83,84,85,86].A reimagined framework for global health that prioritizes health system integration across Universal Health Coverage and Global Health System holistic domains, innovative and unified health financing, cross-sector resilience indicators, and equity as core values offers a necessary path ahead [87,88,89].Resilience in health systems should not be seen as an apolitical outcome, synonymous with a strong health system or improved population health. The resilient healthcare systems should be assessed by considering how macro-level and meso-level structures support adaptive capacity at the micro-level [90,91,92].Leadership development must be an established and integrated component of organizations. The gaps were embedded among complex interactions between the health systems’ building blocks. Understanding the complexities in fostering whole-system strengthening through a holistic measure and applying system thinking is crucial [80,85,89],Democratic rights, human rights, equity, and ethics values have become prominent in national policy debates. Increased attention to an active community voice and the resources it requires can help practitioners to achieve improved health outcomes and researchers to understand the pathways to health improvement through collaboration [81,93]Democratic accountability and the rule of law cannot be completely suspended in any public health emergency. As humanity grapples with COVID-19, the way to combat one pandemic is not to create another, a pandemic of authoritarian rule. Reimagining a narrative should put the collective health and human rights at the center [88,94].Establishing the scope of public health is crucial to understanding its role within the larger health system and adds to the discourse around the relationship between public health, healthcare, and population health. Digital transformation must be further driven to ensure collaboration. Holistic, preventive policies must be adapted to local contexts and implemented through strong local health systems able to have the capacity to respond to emergencies [85,90,95,96].

The findings indicate the comprehensive proposals for redesigning the healthcare system, with the most focus on integrative, holistic, and person-centered approaches.

## 4. Discussion

In this section, the authors discuss the obtained results and interpretations from the perspective of previous studies, policy-planning documents, and statistics. The analysis of scientific literature, content analysis, and statistical data analysis were used to achieve the aim of this study. Regarding the weaknesses of this study, the statistical data from one country were collected, and although Latvia is a relatively small country, it is a statistically reliable sample (at least in the scope of the Baltic region) that can be used to draw attention to the most critical issues.

The literature review on the probable causal interrelation between COVID-19 and ME/CFS resulted in considerations regarding the similarities between post-acute COVID-19 symptoms and ME/CFS, with a lack of sufficient evidence to establish SARS-CoV-2 as an infectious trigger for ME/CFS. At the same time, it was noted that the post-acute sequelae of SARS-CoV-2 infection (PASC) includes a post-coronavirus disease 2019 (COVID-19) syndrome (‘long-COVID’) or features that can follow other acute infectious diseases and ME/CFS. Similarities were observed when comparing the International Consensus Criteria for the diagnosis of ME with the symptoms described for long-COVID-19. The following are suggestions for additional research on the molecular mechanisms that might explain the fatigue and related symptoms in both illnesses: additional patient investigations with longer follow-ups, more uniform criteria for ME/CFS diagnosis, including both in- and out-patients with infections of different severities and a control group, future research to identify the underlying mechanisms, the development of standardized diagnostic criteria, and the establishment of therapies to prevent and treat fatigue and cognitive impairment.

Some findings are focused on the necessity of creating a case definition of what is collectively termed ‘post-viral syndrome’, noting that long-COVID research may also be applicable to a wider range of chronic illnesses, including ME/CFS, which similarly lack adequate understanding and recognition. In light of the prevalence issues, the most critical preliminary findings raise concern regarding a possible future ME/CFS-like pandemic in COVID-19 survivors.

In the authors’ view, regardless of the new challenges of nosology, there is an obvious tendency towards an increase in the burden of complex chronic diseases of unclear etiology. Under these circumstances, the importance of precision medicine is growing, including the research and use of biomarkers for diagnostics. At the same time, the context in which biomarkers are used may change; for instance, ME/CFS may move from the rare disease category to a wide group of diseases with similar symptoms. Accordingly, the emphasis may shift from the use of biomarkers to ‘identify ME/CFS patients’ to the new role of ‘distinguishing ME/CFS patients’ in a group of symptomatically similar illnesses. Certainly, this is valuable if there are specific distinguished treatments provided.

Another consequence of the COVID-19 pandemic is changes in mortality rates in indirect COVID-19-related diagnoses. The example of Latvia demonstrates that during the pandemic, mortality notably increased in the group of diseases of the circulatory system (I00-I99). It can be assumed that this tendency would be associated with significant restrictions on primary healthcare availability during the pandemic, but in-depth research would be needed to identify the impact factors.

Concerning vaccination, it may seem positive that precision medicine’s medicinal products became widely available and used throughout the population. However, the result is ambiguous: with particularly intensive and global-scale use and the oncoming of new resistance, ‘vaccine resistance’ is observed, which can rapidly expand the scope of existing ‘antibiotic resistance’ and its induced problems, posing significant challenges to contemporary society.

Developments in recent months (also illustrated by the example of Latvia, Figure 2) do not confirm the likelihood of achieving the goal set in the WHO’s ’Strategy to Achieve Global COVID-19 Vaccination by mid-2022′, which declared the urgent actions required by the global community to vaccinate 70% of the world’s population against COVID-19 by mid-2022 in order to substantially increase population immunity globally to protect people everywhere from disease, protect the health system, fully restart economies, restore the health of society, and lower the risk of new variants [97]. Presumably, the tactic needs to be changed to maintain the strategic goals.

Concerning healthcare organization and governance, an analysis of the literature was carried out to explore the conceptual issue by investigating the scientific publications on healthcare governance over the past two years. The publications were found to be prevalently dedicated to the non-congruence of the existing system and revealed the need to shift from hierarchical structural governance principles to a people-centered approach. Findings suggest that the pandemic has highlighted the shortcomings of the dominant hierarchical model of healthcare governance when decisions on medical manipulation and resource spending are made at the highest levels of governance, detached from the perspective of medical professionals and the strategic directions of medicine’s development. One of the hallmarks of the current era is that medicine, epidemiology, and healthcare management methods are evolving in parallel, but each at its own pace, and it can be assumed that molecular biology and biomedicine develop more broadly and faster than epidemiology and healthcare management methods. The strengths of this research are based on the investigation of the current trends to pay attention to possible expected and unexpected consequences of the COVID-19 pandemic.

Researchers emphasize the need for new models of healthcare governance. Therefore, further discussions and research could focus on how a holistic approach to patients’ healthcare, treatment decision making, and financial management can be achieved, with a view towards collaborative, inclusive, and participative practices for healthcare professionals and patients, moving from fragmentation to adaptive self-organization, thereby creating well-integrated, equitable, and prosperous societies.

## 5. Conclusions

The circumstances of the crisis distinctly reveal whether and how policy initiatives perform in the real world. The COVID-19 pandemic also revealed some consistencies and inconsistencies in healthcare and accelerated certain processes in society. The outcomes of this study indicate possible future challenges for healthcare and society as a whole, such as an increase in the burden of complex chronic diseases of unclear etiology, ’vaccine resistance’ in addition to existing ‘antibiotic resistance’, and a need for changes in healthcare governance. The results can help in forecasting future scenarios, to increase their value under persistent conditions of uncertainty. This study has outlined the contours for further research.

## Figures and Tables

**Figure 1 healthcare-10-01018-f001:**
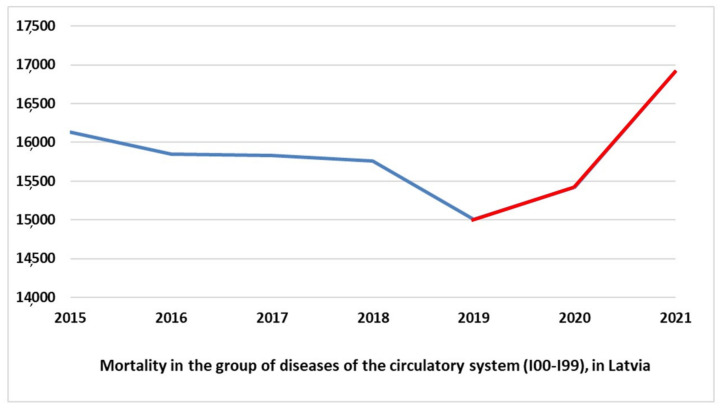
Mortality in the group of diseases of the circulatory system (I00-I99), in Latvia, 2015–2021 (constructed by the authors, using the publicly available statistical data of the Latvian Centre for Disease Prevention and Control (CDPC) [51]).

**Figure 2 healthcare-10-01018-f002:**
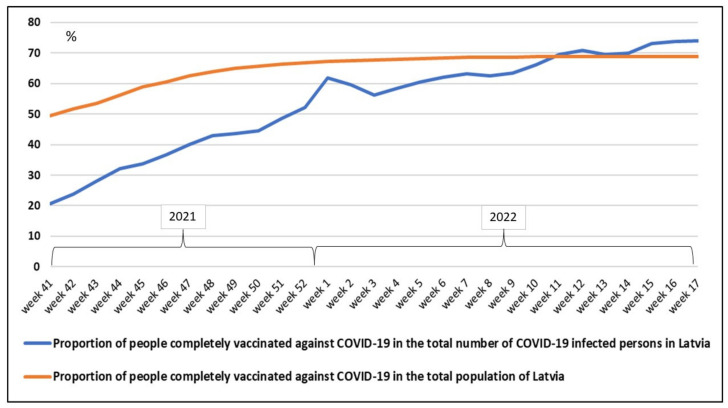
Proportion of people completely vaccinated against COVID-19 in the total population of Latvia, and proportion of people completely vaccinated against COVID-19 in the total number of COVID-19-infected persons in Latvia, in a period from week 41 of 2021 until week 17 of 2022 (constructed by the authors, using the statistical data of the competent authorities of Latvia [74,75]).

## Data Availability

The data supporting reported results are available on websites: https://statistika.spkc.gov.lv/pxweb/en/Health/Health__Mirstiba/MOR13_celoni_vecgrupas_COVID-vakcinacija.px/ (accessed on 5 April 2022); https://stat.gov.lv/en/statistics-themes/population/population-number/247-population-number-its-changes-and-density?themeCode=IR (accessed on 5 April 2022); https://covid19.gov.lv/covid-19/covid-19-statistika/covid-19-izplatiba-latvija (accessed on 5 April 2022).

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
