# Peer review of "COVID-19 Pandemic-Revealed Consistencies and Inconsistencies in Healthcare: A Medical and Organizational View"

_healthcare, 2022, doi:10.3390/healthcare10061018_

Round 1
Reviewer 1 Report
The work entitled "COVID-19 pandemic-revealed consistencies and inconsistencies in healthcare: medical and organisational view" provides significant proof of the COVID-19 pandemic consistencies and pandemic-revealed inconsistencies in healthcare. The work is extremely well done and organized. The methodology employed to conduct this review work is very detailed and precise, which allows readers to understand very quickly the goal of the authors. Most importantly, there is novelty in thies work, since such a data (to this Reviewer's knowledge) has never been published. There are some English writing mistakes that must be fixed, but they do not ompromise the comprehension of the work. I recommend taht the authors review the entire manuscript once again for English mistakes so it can be published.
Author Response
Thank you very much for the inspiring valuation.
Please see the attachment.

Reviewer 2 Report
ID: healthcare-1698128
Title: COVID-19 pandemic-revealed consistencies and inconsistencies in healthcare: medical and organisational view.
Thank you for the opportunity to review this paper. Here are some comments for you.
Comment:
Minor revision.
Detailed information:
Abstract
Page 1, line 8-15: It is not advisable for the background section to take up half of the abstract. Refine the most important research background with more concise words. Moreover, “COVID-19” is not a good word for readers who firstly read this kind of article. You’d better add official full name of WHO before it.
Materials and Methods
Page 4: I think the structure of this section is a bit confusing. You'd better write it more logically. Also, you don't need to write where these research methods correspond to the results.
Page 4, line 163: We usually write the full name first and add the abbreviation after the full name.
Introduction
Page 1-3: I can't grasp the research focus in the introduction, can you highlight your research purpose and hypothesis more?
Results
Page 4, line 174-175: Is this sentence really necessary?
Page 4, line 178-179: Why should NE and CFS be chosen as representative diseases? What is the prevalence of these two diseases? Are there any previous studies on this?
Page 5, Figure 2: This figure does not look very clear.
Page 6, Table 1: To make it easier for readers, you can list the main results of these articles with key points, or summarize the similarities and differences.
Page 7, line 260-261: What’s “UNICEF” and “TRANSVAC”? Explain them please.
Page 9, line 317-319: No need to present reasons of analysis in the “results” section, that is the job for the “method” part. Describe the results of your study.
Discussion
Page 11: What are the strengths and weaknesses of this study? What are the future research directions?
Please take these comments into your consideration. This research is interesting, but some of the content is redundant and will make the reader doubt the focus of the study. You can write with detailed key points. Moreover, the logic of the article also needs to be strengthened. In addition, the quality of written English could be improved by consulting a native English speaker.
My best,
Your reviewer
Author Response
Thank you for your valuable comments.
Please see the attachment.

Reviewer 3 Report
This disjointed manuscript discusses healthcare systems in terms of organization, a movement towards precision medicine, and the availability of universal healthcare. It also discusses issues related to post-COVID-19 viral syndromes (paying particular attention to the authors’ previous work on ME/CFS) expanding upon the authors’ previous review. A bullet point list from a literature review of COVID-19 is also presented.
The focus of the introduction seems to be on the theoretical implications of the implementation of universal healthcare in a broad sense. Despite the focus of the abstract being “COVID-19 pandemic consistencies and also to the pandemic-revealed inconsistencies in healthcare,” COVID-19 is only discussed tangentially in the introduction.
It is unclear what is being presented in Figure 1. If included, the figure should be revised to add clarity to the issues around budget constraints related to number of patients treated. The figure caption also needs to be revised to sufficiently explain the figure.
Lines 146-154 – It is unclear what databases were searched, how articles were identified and how they were screened for inclusion. Please revise so that it is clear how this literature search and review was conducted, and the inclusive dates of the search. It might be more appropriate to include these dates in the literature search described in the following paragraph, which would probably identify the same articles, rather than including them due to their use in the self-cited reference [11].
Lines 165-168 – Please describe specific databases used to gather these data and the searches used to produce them.
Lines 169-172 – The intended meaning of this paragraph is unclear.
Figure 2 & methods –18,491 articles identified in the literature search were excluded because they were “marked ineligible by automation tools.” A description of these automation tools and their application is needed in the methods. The exclusion criteria of “reports not retrieved” needs additional explanation. A description of how the decision to exclude 50 references identified in the literature search because they were “non-relevant to research theme or items” is needed.
Lines 193-196 – This seems to suggest that there are 8 ongoing reviews currently submitted for publication and 5 non-published recent reviews covering this same topic. If so, how is this review non-duplicative of these?
Lines 212-219 – This paragraph cites 20 original scientific reports but does not cover enough their content in sufficient depth that it is in anyway informative to the reader. It would be better to review these articles in sufficient detail so that they would be informative to the reader on the scope of current research into this topic.
Lines 252-256 – A claim that there are increases in Diseases of the circulatory system (I00-I99, ICD-10) in Latvia during COVID-19 are not supported with the presentation of data from the Latvian CDPC or citation of other published literature.
Lines 364-406 – it is highly unusual to present the result of the literature review as a bullet point list rather than as cohesive paragraphs. Citing all 17 of the contributing literature sources on a single line (368) instead of referencing them as their ideas are covered is not appropriate and needs to be rectified.
Minor comments
The first paragraph of the introduction (line 28-39) is difficult to follow and too generalized. It is not needed in this review.
There are numerous grammatical errors (especially run-on sentences) throughout the manuscript.
Line 78-79 – citation needed
Line 79-82 – this sentence has some ambiguity. It is unclear what financial resources (or what the source of the expanded coverage and reduced direct payment care) are being discussed in this context.
Line 84-86 – This sentence is difficult to interpret. It is unclear who is referred to in “their objectives”. The factors included in “each of the dimensions” are also unclear. This statement seems to oversimplify a complex concept of the implementation of universal healthcare requiring input and compromise from many stakeholder groups (e.g. government, populace, medical systems, etc.) involving a vast number of factors related to patient care coverage, costs, and availability.
Lines 87-90 – citations are needed for the claims on the evolution of medicine, epidemiology and healthcare management.
Lines 94-96 – It is difficult to support the claim that mass vaccination is a form of precision medicine.
Lines 122-127 – citations are needed.
Section 3.1.3 provides a nice overview of immunization tendencies (particularly in Europe), but it is unclear how it fits in with the rest of the review.
Figure 5 maybe an oversimplification. The author’s view for a “people-centered healthcare system model” is not well described. It is difficult for the reader to see the point of this figure.
Author Response

(The authors gave the same response as above.)

Round 2
Reviewer 3 Report
The authors have made significant progress on this manuscript which discusses issues related to post-COVID-19 viral syndromes including implications for healthcare systems. However, I still have concerns about the revised manuscript.
Line 79-80: It is unclear how the authors support their claim that “the COVID-19 vaccines which represent the precision medicine's advanced medicinal products,” are in any way related to precision medicine. COVID-19 vaccines are not an example of precision medicine. If the vaccine dose and/or formulation was varied based on an individual’s phenotype (lifestyle and personal characteristics) and genetic background (genotype), then the vaccines would be an example of precision medicine. As they have been implemented in all countries, COVID-19 vaccines are mass vaccination campaigns not precision medicine. Differing vaccine guidance by country or broad age ranges also does not meet the criteria needed to be precision medicine.
Line 84: This needs to be expanded upon so that the reader understands what the authors are referring to as “the current healthcare organizational model.”
Line 91-94: It is unclear if this budget expansion is temporary in response to COVID-19 or a permanent increase in spending to strengthen the health system. It is difficult to comment on this as (lines 98-99 state) “resources allocation in the pandemic period was not evaluated.”
Line 113-122: It appears the authors have chosen ME/CFS as representative diseases for their research based on their previous publication (citation #11). There is a tremendous amount of literature being published currently on post-COVID conditions (also known as Long COVID or post-COVID-19 syndrome) with symptoms that are like those reported by people with ME/CFS. The authors need to clearly delineate how they are distinguishing post-COVID conditions from ME/CFS (if they are) and at least comment on the difficulty in distinguishing the diagnoses for ME/CFS from post-COVID conditions (if data on these conditions is being collected separately).
Line 128-129: Communicating the methods for a literature review search needs to include the keywords searched and the date the search was performed. The inclusion of keywords is important as the search for “the lasts finding regarding the reciprocity of COVID-19 and ME/CFS and health care organizational issues (lines 127-128),” are so broad as to be unintelligible. Additionally, the databases searched must be listed, and because of this, listing “…and other Medical Databases,” is not appropriate.
Lines 130-135: It should be clear from the methods what data was mined from these databases and how it was generated. For the purposes of reproducibility, a reader should be able to recreate the dataset from the listed databases by following the methods. A similar comment was made in my previous review. If the exact same search from the self-cited citation #11 were used, then this work may be “self-plagiarism” of that article as it would likely include the same results and analysis of these searches.
Lines 145-151: In-text citations are needed to support these claims despite the information (with citations) being included in Table 1.
Line 160: Which “short communication and viewpoint articles” are the authors referring to here?
Line 168-170 is unclear which “conditions for changes in mortality rates, which are unlikely to be related to COVID-19 infection” are being discussed.
Figure 1 and related discussion on disease of the circulatory system linked to COVID-19 seems to give the impression that this is all mortality not related to COVID-19. However, COVID-19 has been linked to diseases of the circulatory system. There is a significant volume of published literature on the increase in excess mortality or other conditions that are related to COVID-19 despite not being included in COVID-19 related morbidity and mortality statistics.
Line 23 and 274: Vaccine-resistance can be a confusing term in that it can apply to both vaccine hesitancy and vaccine resistant viral variants. In this case, it refers to the latter. Please consider using vaccine resistant viral variants instead of vaccine-resistance.
Line 273-275: The emergence of vaccine resistant viral variants has been thoroughly explored in the published scientific literature. Some of that literature should be cited as part of this discussion as this is not a new phenomenon reported in this manuscript.
Line 311: Again, the details of the way “the scientific literature was analyzed” are needed in the methods.
Line 319-320: Because the methods of the literature search are vague, it is unclear if, “the literature review indicated the lack of original articles dedicated to health systems construction and governance during a pandemic,” or if these articles were not found with the keywords searched.
While the authors have reclassified the manuscript as a research “article”, it still reads more like a literature review on the topics discussed. As such, the previous reviewer issues with Figure 2 were addressed by deleting this figure which outlined the literature review methods. It would be more appropriate to retain this figure and address the previous reviewer comments on it.
There is an error in the bibliography where item 10 is followed by four items all labeled “1.”
Author Response
Thank you for the valuable suggestions.
Please see the attachment.
